# Are we using Motion in Referring Segmentation? A Motion-Centric Evaluation

## Abstract

Multi-modal large language models (MLLMs) have shown impressive generalization across tasks using images and text modalities. While their extension to video modality has enabled tasks such as video question answering and video captioning, their dense spatiotemporal understanding, particularly in referring video segmentation, is less studied. In this work, we raise the pertinent question of whether motion is used in referring segmentation and whether video MLLMs designed for this task truly leverage motion cues when segmenting objects based on natural language expressions. We identify shortcomings in the current benchmarks, where we show that a single frame can often suffice for capturing the motion referring expression without any temporal reasoning. To address this, we introduce a motion-centric probing and evaluation framework that automatically selects keyframes within videos designed to mislead models with apparent motion lacking true spatiotemporal change. This is used to assess whether models rely on genuine motion cues or merely static visual features. Our empirical analysis reveals that existing video MLLMs underutilize motion information in this dense prediction task. It also shows the kind of properties existing in referring expressions that make it more motion-oriented than others. We further establish strong baselines using MLLMs that outperform prior methods, offering new insights into the interplay between spatial and temporal information in dense video-language understanding tasks. Our motion-centric evaluation and findings challenge future models to improve dense spatiotemporal grounding and pixel-level understanding within videos.

## 1 Introduction

Multi-modal large language models (MLLMs) have recently emerged as general-purpose tools that can operate on input image/video and text Liu et al. (2023); Bai et al. (2023); Wang et al. (2024); Bai et al. (2025); Liu et al. (2024); Zhu et al. (2025). They can be language guided through instructions in addition to various visual prompting techniques to produce the desired output. They extend powerful large language models trained within an autoregressive modelling framework, coupled with pre-trained vision encoders, vision-language alignment and projectors. The extension of multi-modal large language models to operate on videos has been extensively investigated Lin et al. (2023); Maaz et al. (2023); Fu et al. (2024); Bai et al. (2025); Zohar et al. (2024), yet it mainly focused on coarse output, such as its use in video question answering, video captioning or video-level grounding. However, few methods focused on dense spatiotemporal output such as pixel-level visual grounding in videos, also referred to as referring video segmentation Yan et al. (2024); Munasinghe et al. (2024).

Video segmentation generally focuses on identifying different segments in a video that can be defined based on semantics, saliency or language guided Zhou et al. (2022). The latter is the main focus of this work, where models aim to segment objects of interest in videos based on a referring expression. Referring video segmentation emerged with the introduction of A2D sentences Gavrilyuk et al. (2018)

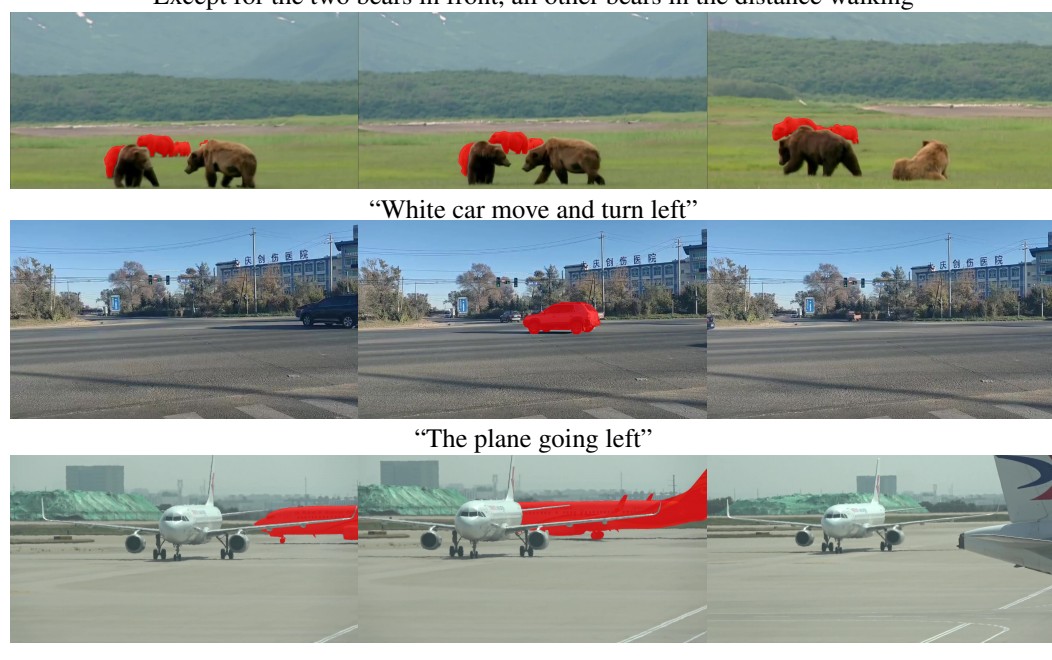

"Except for the two bears in front, all other bears in the distance walking"

"White car move and turn left"

"The plane going left"

(a)         (b)         (c)

Figure 1: Motivation behind our research question, where we show that motion referring expressions can still be captured from a single image (i.e., middle column) referred to as the keyframe, without the need of dynamic information. It shows three videos from MeVIS benchmark Ding et al. (2023), (a) first frame, (b) keyframe that best captures the expression and (c) the last frame. Ground-truth segmentation highlighted in red. It shows the color, heading, position or category as sufficient cues to identify the objects without motion.

and referring DAVIS'17 Pont-Tuset et al. (2017). It is the counterpart task of referring segmentation in images Kazemzadeh et al. (2014); Yu et al. (2016) extended to videos with the added challenges in ensuring the temporal consistency of the segmentation across the video and identifying motion referring expressions Ding et al. (2023). Initial methods relied on the advancements in transformer-based architectures and masked modelling Wu et al. (2022), followed by multi-modal large language models with autoregressive modelling Yan et al. (2024); Munasinghe et al. (2024). Concurrent to the aforementioned developments, the benchmarks designed to evaluate these methods were improved to push the boundaries on the task towards better reasoning Yan et al. (2024) and an understanding of motion Ding et al. (2023).

In this work, we ask the major question: "Is Motion properly used in video referring segmentation techniques?", revisiting the temporal understanding in these models beyond what was provided in earlier works. While previous interpretability works in video segmentation showed consistent failures in utilizing dynamic information Kowal et al. (2022, 2024), yet they were not language guided and were constrained to precursor methods to MLLMs. Other works in interpretability and benchmarking have studied a similar question in video language modelling but they were not designed for dense spatiotemporal grounding and were rather confined to coarse video question answering tasks Buch et al. (2022). The dense spatiotemporal segmentation task within videos makes it more interesting to study the ability of video MLLMs in capturing temporal information. Since models can be deceived to use spatial information only, leaving out the temporal information, or they can easily rely on coarse temporal information without a proper understanding of the full spatiotemporal dynamics within the video. As such, we provide a study of the ability of video MLLMs in utilizing motion information within language-guided video segmentation. Towards the latter, we create a motion-centric probing and evaluation technique that questions previous efforts on motion referring expressions segmentation and can be used to drive better understanding of video MLLMs performance.

Figure 1 presents the motivation of our work, as it shows the shortcomings in motion referring video segmentation benchmarks when considering motion. We show three examples with the first frame, keyframe and the last frame from each video with the respective referring expression. It clearly shows

that the keyframe (middle column) can be sufficient to understand which object is referred to in the expression from one static frame without the need for temporal information or a proper understanding of dynamics. Consequently, we propose an automatic mechanism to select such keyframes and use them to generate a motion-centric benchmark, challenging video MLLMs to differentiate motion from what appears to be fake motion.

In summary, our contributions include: (i) An empirical analysis of the shortcomings of video MLLMs in utilizing temporal information within such a dense prediction task, (ii) proposing a motion-centric probing and evaluation technique for referring video segmentation using a simple keyframe automatic selection, and (iii) providing strong baselines using Video MLLMs that outperform the state-of-the-art methods and allow to study the use of spatial information from a single image vs. coarse temporal information in such a challenging setup.

## 2   Related Work

**Multi-modal large language models (MLLMs) and benchmarking.** Pioneering works in multi-modal large language models such as LLaVA Liu et al. (2023/, 2024), Cambrian-1 Tong et al. (2024), Qwen-VL Bai et al. (2023) and InternVL Chen et al. (2024b) have driven significant development towards the creation of general-purpose agents. Consequent works that built upon these developments to equip MLLMs with better spatial and temporal understanding emerged Wang et al. (2024); Bai et al. (2025); Chen et al. (2024a); Zhu et al. (2025); Lai et al. (2024); Rasheed et al. (2024); Zhang et al. (2024a,b); Munasinghe et al. (2024); Yan et al. (2024); Zohar et al. (2024). Some of these methods have the capability to perform visual grounding in either images or videos on the region-level or pixel-level Lai et al. (2024); Rasheed et al. (2024); Zhang et al. (2024a,b); Munasinghe et al. (2024); Yan et al. (2024). Other works focused on extending to video MLLMs Bai et al. (2025); Munasinghe et al. (2024); Yan et al. (2024); Zohar et al. (2024). One of the major drivers behind these developments is the evaluation benchmarks that push the limit on these models and ensure improved performance, in addition to studies that interpret their behaviour.

There is an abundance of standard benchmarks used to evaluate MLLMs (e.g., MMU Yue et al. (2024)) and pixel-level benchmarks (e.g., refCOCO/+/g Yu et al. (2016); Kazemzadeh et al. (2014)). Moreover, benchmarks designed to evaluate video MLLMs have emerged, such as MMBench-Video Fang et al. (2024) and Video-MME Fu et al. (2024). Concurrent work studying video MLLMs Zohar et al. (2024) has shown the bias within such evaluation benchmarks towards using a single image or textual input only instead of fully evaluating the use of temporal information. Nonetheless, the majority of previous works on video MLLMs benchmarking and analysis focused on coarse output, specifically with video question answering. While there are recent benchmarks evaluating pixel-level visual grounding in videos Yan et al. (2024); Ding et al. (2023), we show that they are ineffective in assessing the ability of video MLLMs to capture dynamics. As such, we propose a novel probing technique that is motion-centric and independent of the benchmark. It is coupled with a pixel-level visual grounding benchmark towards a motion-centric evaluation. Using this probing technique enables a better understanding of video MLLMs and their ability in dense spatiotemporal understanding, which goes beyond coarse tasks such as visual question answering.

**Video segmentation.** The general task of video segmentation that takes as input a video clip and outputs segments within the video based on the definition for the objects of interest that can either be: (i) based on semantic categories (i.e., video semantic segmentation) Miao et al. (2021), (ii) within foreground/background segmentation framework relying on saliency or tracking (i.e., video object segmentation) Karim et al. (2023), (iii) based on language (i.e., referring video segmentation) Munasinghe et al. (2024); Yan et al. (2024); Wu et al. (2022); Ding et al. (2023). We focus on referring video segmentation that relies on a referring expression describing the object/s of interest to be segmented within an input video. Early methods for referring video segmentation relied on masked modelling from RoBERTa Wu et al. (2022); Ding et al. (2023). However, with the MLLMs development, better referring video segmentation models were developed that relied on these autoregressive models with image/video and text input Munasinghe et al. (2024); Yan et al. (2024). A recent method extended the referring video segmentation task to the more challenging video reasoning and segmentation task Yan et al. (2024).

Another track of methods focused on emphasizing motion in referring video segmentation Ding et al. (2023). However, our work shows the shortcomings in the aforementioned benchmarks and

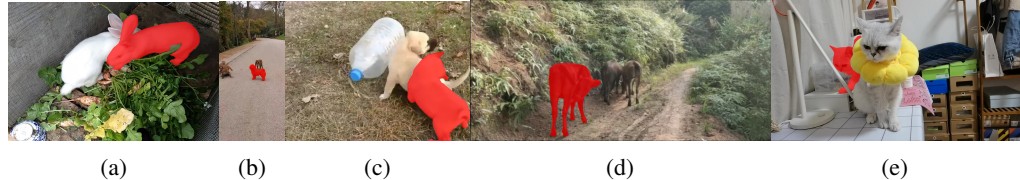

(a)  (b)  (c)  (d)  (e)

Figure 2: Qualitative analysis of our proposed automatic keyframe selection from five examples that show a single frame can be sufficient to convey the motion expression without any motion involved. It is mainly conveyed through the use of static cues such as the heading, object type, or position. Expressions of each example are as follows: (a) "jump to the left then jump back", (b) "dog playing with monkey", (c) "puppy that overwhelms another puppy", (d) "cow shaking head and looking at us." (e) "The little cat walking from behind to the front".

evaluations, where we show that state-of-the-art methods and our proposed baselines that surpass them, all fail in our motion-centric evaluation. Since our probing can deceive models into believing there is a depiction of the motion expression when it is only a single static frame. While recent interpretability studies looked at video segmentation models and their ability to capture dynamic information, they have mainly focused on methods that are not language guided, unlike ours Kowal et al. (2022, 2024); Karim et al. (2023).

## 3 Method

In this section, we summarize the shortcomings in the current referring video segmentation methods, including the ones that rely on multi-modal large language models. Then we describe our motion-centric probing, the respective benchmark and our proposed strong baselines that assess the use of spatiotemporal information.

### 3.1 Shortcomings in Video MLLMs and Referring Video Segmentation Benchmarks

While there have been previous works focusing on establishing single-image baselines for video-language understanding on coarse-level tasks Buch et al. (2022), (e.g., video question answering or video-language retrieval), we are the first to explore this within dense spatiotemporal tasks. We focus on referring video segmentation and put emphasis on both standard and motion referring expressions Ding et al. (2023). We argue that the majority of referring expressions can be identified using strong single-image baselines that do not have an understanding of temporal information. While motion referring expressions may seemingly use motion, we show that such expressions can still be identified from one static frame. Figure 2 shows five examples that can be sufficiently identified with static information without the use of temporal information, thus showing a weakness in the current evaluation benchmarks.

There are three main properties in the motion referring expressions benchmark introduced to evaluate the ability of models in capturing motion, these include: (i) the selection of video content that contains multiple objects that coexist with motion, (ii) prioritizing referring expressions that do not contain static cues such as color, and (iii) the use of multi-object interactions. Except, we argue that these properties, standalone, are insufficient. While the selection of videos that have multiple objects coexisting can be differentiated through their actions or motion, the majority of these actions or motions can be inferred from one frame. Figure 2a,2d,2e show examples that can be identified from one frame, where the direction, heading or the object's current position within the image from the referred expression can be deducted from one static frame. The second property is impractical, since the identification of the object category can be a significant static cue already, and in other scenarios, the only way to differentiate objects is through color. Thus, it is infeasible to fully decouple the static cues from the motion information (e.g., Fig. 2b with only one dog in the scene). Finally, the interactions between multiple objects can indeed be a drive for evaluating motion, but in certain scenarios it can be inferred from a single frame (e.g., Fig. 2c where the expression describing the two objects' interaction can be identified from a single image).

As such, we propose a motion-centric probing and evaluation to study the shortcomings in video MLLMs with respect to capturing spatiotemporal dynamics. Additionally, we propose quantitative

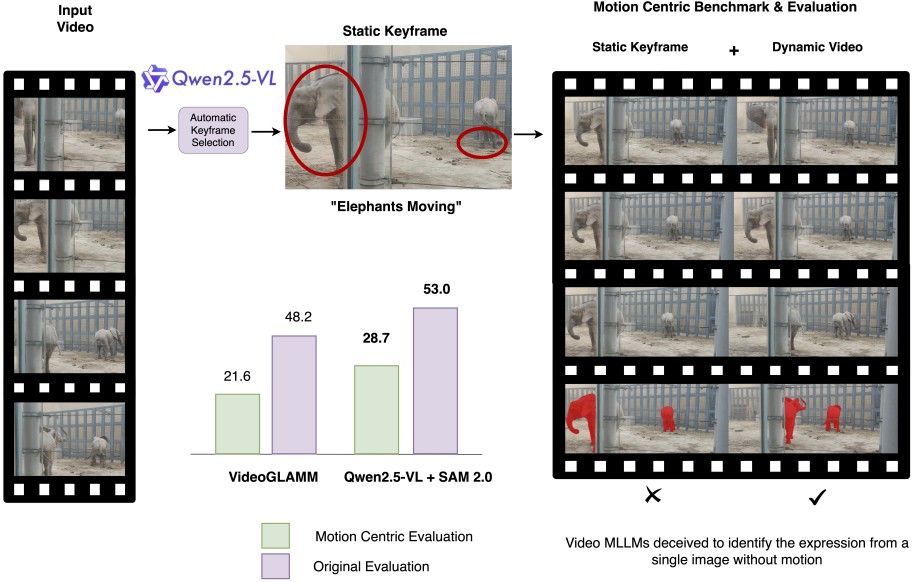

Figure 3: Detailed mechanism for our motion-centric probing that relies on the automatic selection of a static keyframe to mislead MLLMs into an existent motion (highlighted in red circles), when there is no motion involved. The output video combining the original dynamic video and the static keyframe is used to evaluate the video MLLMs ability to differentiate between true motion vs. deceiving ones. The last frame shows the predictions from our strongest baseline (i.e., Qwen2.5-VL + SAM 2.0†) highlighted in red. Even with such a strong video MLLM baseline, it is still misled to believing an existing motion when there is not. It also shows that the true performance of state-of-the-art video MLLMs (VideoGLAMM and Qwen2.5-VL + SAM 2.0†) is downgraded by half when using the motion-centric evaluation vs. the original evaluation on the *mini-validation* set on MeVIS dataset.

means to analyze the properties existing in referring expressions that can drive better evaluation for the use of motion beyond a single image.

## 3.2 A Motion-Centric Benchmark

**Benchmark.** Figure 3 describes our motion-centric probing. First, we identify the keyframe in the video that can approximate the motion expression with one static frame. Towards that, we use a multi-modal large language model that has the capability of coarse video grounding, i.e., identifying the frames temporally that correspond to a certain expression without identification of the object spatially within the frames. In our case, we use Qwen2.5-VL and prompt it to identify the expression using the following: `Given the query: <EXP>, when does the described content occur in the video? Output the first and last seconds for this action in JSON format.` The output is further processed to identify the temporal window of frames with the middle frame labelled as the keyframe capturing this motion. Figure 2 shows five example keyframes selected with their motion referring expression. We use the retrieved keyframes to create a video containing both the static keyframe in addition to the original video clip as shown in Figure 3.

**Strong baselines.** We establish strong single-image and semi-temporal baselines for referring video segmentation using powerful multi-modal large language models that can visually ground objects on the region level. Models such as Qwen2.5-VL Bai et al. (2025) and InternVL3 Zhu et al. (2025) have emerged that show strong capabilities in visually grounding objects and outputting their corresponding bounding boxes. We build three baselines that are biased to the static information conveyed from a single image. The first baseline relies on the MLLM output, followed by using the segment anything model Kirillov et al. (2023); Ravi et al. (2024) to generate the output segmentation per frame for the corresponding referring expression (MLLM + SAM). We use the following prompt: `Locate the <EXP>, output its bbox coordinates using JSON format.` The second baseline follows a similar procedure but identifies the object in the first frame, then it uses the segment anything model 2.0 Ravi et al. (2024) for tracking throughout the video, (MLLM + SAM 2.0). The final baseline

| Method | Backbone/Base MLLM | RefDAVIS-17 | | | MeVIS | | |
|---|---|---|---|---|---|---|---|
| | | $\mathcal{J}$ | $\mathcal{F}$ | $\mathcal{J}\&\mathcal{F}$ | $\mathcal{J}$ | $\mathcal{F}$ | $\mathcal{J}\&\mathcal{F}$ |
| ReferFormer | VideoSwin-B | 58.1 | 64.1 | 61.1 | 29.8 | 32.2 | 31.0 |
| LMPM | Swin-T | - | - | - | 34.2 | 40.2 | 37.2 |
| LISA | LLaVA-7B | 62.2 | 67.3 | 64.8 | 35.1 | 39.4 | 37.2 |
| VISA | Chat-UniVi-7B | 66.3 | 72.5 | 69.4 | 40.7 | 46.3 | 43.5 |
| VideoGLAMM | VEn.+Phi3-Mini-3.8B | 65.6 | 73.3 | 69.5 | 42.1 | 48.2 | 45.2 |
| MLLM + SAM | Qwen2.5-VL-7B | 60.8 | 66.8 | 63.8 | 38.8 | 45.0 | 41.9 |
| MLLM + SAM 2.0 | Qwen2.5-VL-7B | **69.9** | 66.5 | 68.2 | 40.9 | 47.4 | 44.2 |
| MLLM + SAM 2.0† | Qwen2.5-VL-7B | 69.8 | **75.9** | **72.9** | **44.5** | **51.2** | **47.8** |

Table 1: Comparison of our strong single-image and semi-temporal baselines (last three rows) with respect to the state-of-the-art methods on RefDAVIS'17 and MeVIS datasets. Our baselines are mostly biased to the static information, yet surpass previous methods. VEn.: vision encoders for CLIP and InternVideov2 used in VideoGLAMM. †: indicates the use of the automatic keyframe selection. Best results are bolded.

relies on identifying the keyframe in the input video that can have the referred expression, then it uses that as the initialization frame for SAM 2.0 (MLLM + SAM 2.0†). The keyframe is retrieved following the previous method in the motion-centric probing. The last baseline can be looked upon as a semi-temporal baseline, since it is an intermediate baseline between full spatiotemporal and coarse temporal grounding. It identifies the object within the video on the coarse level and for the pixel-level grounding it only uses one static keyframe, followed by propagating the segmentation in the video.

# 4 Experimental Results

## 4.1 Experimental Setup

**Evaluation datasets and metrics.** We use established referring video segmentation datasets in our evaluation including RefDAVIS17 Pont-Tuset et al. (2017) and the motion referring expression dataset MeVIS Ding et al. (2023). For RefDAVIS17 we evaluate on the provided *validation* split that includes 30 videos, while for MeVIS, we use two splits: the *mini-validation* subset, which includes 50 videos with their corresponding ground-truth available and the *validation* subset that includes 140 videos that do not have their ground-truth publicly available but can be evaluated upon using MeVIS evaluation server [1]. Finally, we evaluate on our constructed motion-centric version of MeVIS *mini-validation* as detailed in Sec. 3.2. We use the standard evaluation metrics for the region similarity, $\mathcal{J}$, which computes the mean intersection over union, the contour accuracy, $\mathcal{F}$, and the average of both, $\mathcal{J}\&\mathcal{F}$.

**Compared methods.** We compare our strong baselines with respect to state-of-the-art referring video segmentation methods, including the prior ones that relied on masked modelling from the RoBERTa model, such as ReferFormer Wu et al. (2022) and LMPM Ding et al. (2023). Additionally, we compare against recent ones that rely on the power of large language models with autoregressive modeling in LISA Lai et al. (2024), VISA Yan et al. (2024) and VideoGLAMM Munasinghe et al. (2024). For the motion-centric evaluation, we focus specifically on the best models in each category, which have their codes and weights publicly available (i.e., LMPM and VideoGLAMM), in addition to our three strong baselines.

## 4.2 Strong Baselines Evaluation

In this section, we show that our baselines already provide state-of-the-art performance, surpassing previous Lai et al. (2024); Yan et al. (2024) and concurrent Munasinghe et al. (2024) works. Table 1 shows results across two benchmarks, including the motion referring expression segmentation dataset. It clearly shows that the simple baseline, MLLM + SAM 2.0, that does not incorporate any temporal information in the identification of the referred expression, surpasses the state-of-the-art methods on MeVIS except our concurrent work, VideoGLAMM. While our strongest baseline, MLLM + SAM 2.0†, that relies on partial temporal information, outperforms the previous state of the art.

---

[1] https://codalab.lisn.upsaclay.fr/competitions/15094

| Method | Backbone/Base MLLM | MeVIS *mini-validation* | | | MeVIS motion-centric | | |
|---|---|---|---|---|---|---|---|
| | | $\mathcal{J}$ | $\mathcal{F}$ | $\mathcal{J\&F}$ | $\mathcal{J}$ | $\mathcal{F}$ | $\mathcal{J\&F}$ |
| LMPM | Swin-T | 34.2 | 40.2 | 37.2 | 18.2 | 26.9 | 22.6 |
| VideoGLAMM | VEn.+Phi3-Mini-3.8B | 43.6 | 52.7 | 48.2 | 18.1 | 25.1 | 21.6 |
| MLLM + SAM | Qwen2.5-VL-7B | 48.8 | 56.0 | 52.4 | **24.1** | 33.8 | **28.9** |
| MLLM + SAM 2.0 | Qwen2.5-VL-7B | 46.9 | 55.3 | 51.1 | 22.9 | 32.8 | 27.9 |
| MLLM + SAM 2.0† | Qwen2.5-VL-7B | **49.1** | **56.9** | **53.0** | 23.4 | **34.2** | 28.7 |

Table 2: Quantitative comparison of state-of-the-art models and our proposed baselines on the MeVIS *mini-validation* set and our corresponding motion-centric one. VEn.: vision encoders for CLIP and InternVideov2 used in VideoGLAMM. †: indicates the use of the automatic keyframe selection. It clearly shows a strong drop in performance for all the models, including our baselines. Best and second-best results are bolded and underlined, respectively.

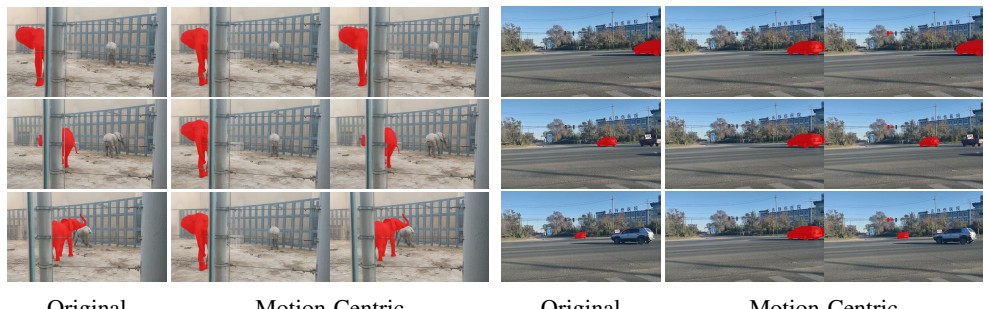

| Original | Motion-Centric | Original | Motion-Centric |

Figure 4: Qualitative analysis comparing the original MeVIS *mini-validation* set performance vs. the motion-centric one that incorporates an additional static keyframe to the video to confuse video MLLMs. Predictions of our strongest baseline, Qwen2.5-VL + SAM 2.0† are highlighted in red. Motion referring expressions for the examples are as follows: (i) first column is "front elephant walking to backwards", (ii) second column is "black car move and turn left".

Thus, it confirms that our baselines, without temporal or with semi-temporal information, establish strong results that we can safely use for our motion-centric probing and evaluation. While our baselines rely on a stronger base multi-modal large language model, they are only meant to motivate the motion-centric evaluation and identify their shortcomings on our proposed benchmark and the corresponding analysis of the referring expressions.

## 4.3 A Motion-Centric Evaluation

In this section, we focus on the motion-centric evaluation, where we show that both the state-of-the-art methods and our strong baselines still fall short in differentiating between the objects in the static keyframe and real motion. Table 2 shows the evaluation on the *mini-validation* set of MeVIS and the corresponding motion-centric version. Across all the methods, there is an obvious decrease of around half the original performance on the standard videos that do not include that static keyframe. It highlights the major shortcoming that the majority of the methods do not have an understanding of the temporal information and are largely still biased to one static frame.

**Qualitative comparison original vs. motion-centric.** Figure 4 shows the qualitative analysis for our strongest baseline, MLLM + SAM 2.0†, which relies on identifying the keyframe and then propagating the information across the video. Even with the strongest baseline, the models tend to segment the objects based on static cues in the referred expression. Hence, in the two examples provided, the "front elephant" and the "black car" got segmented regardless of the motion incurred and without an understanding of the full referring expression.

**Qualitative ablation on motion-centric.** Furthermore, we show a qualitative ablation of our strongest two baselines compared to our concurrent work, VideoGLAMM, in Figure 5 on the motion-centric benchmark. It shows three example sequences with three frames each, where our baseline, Qwen2.5-VL + SAM 2.0†, that uses partial temporal information, not the full spatiotemporal information, has a better ability to differentiate the static frame from the dynamic video than the baseline that does

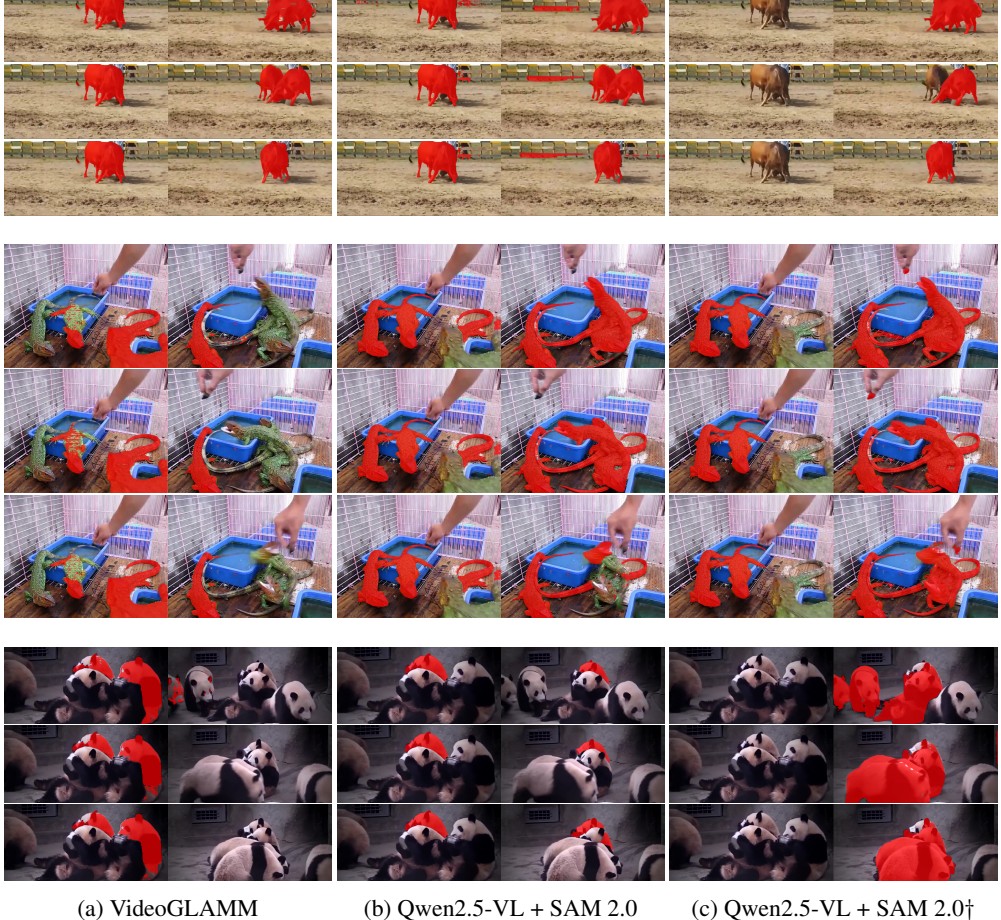

|                   |                       |                        |
|:-----------------:|:---------------------:|:----------------------:|
| (a) VideoGLAMM    | (b) Qwen2.5-VL + SAM 2.0 | (c) Qwen2.5-VL + SAM 2.0† |

Figure 5: Qualitative ablation comparing our two strongest baselines and concurrent work from VideoGLAMM. The motion referring expressions are as follows: (i) The first three rows are "two fighting cows heading each other". (ii) The second three rows are "4 lizards moving around". (iii) The final three rows are "Panda turning around and moving forward from leftmost to rightmost."

not use any temporal information, Qwen2.5-VL + SAM 2.0. This is evident in the following: (i) the first example our strongest baseline does not segment the static frame, (ii) in the second example it wrongly segments only two lizards in the static frame which appear to be about to move and (iii) in the third example it segments the correct bear but also segments false positive ones unlike the static baseline that was biased to the expression "rightmost" or "leftmost". Nonetheless, even with such a strong baseline that outperforms state-of-the-art results, it is quite challenging to differentiate the static from the moving objects, where static cues in the referred expression can be misleading. Note that in our motion-centric probing, there is an additional challenge from the expressions that use location cues, e.g., "leftmost" or "rightmost", as it raises the question of what reference defines these locations. However, models that can capture motion can overcome such a challenge since it is not only using the static cues but rather the motion and full expression describing it.

**Analysis on the referring motion expressions.** In order to study the type of referred expression that can be misleading beyond the visual contextual information, we use our strongest baseline, Qwen2.5-VL + SAM 2.0†, and compute the false positives in the static frame within our motion-centric evaluation per video and referred expression. Specifically, we compute the ratio of the segmented area in the static frame with respect to the full area of the image and group the referred expressions into two major groups: the ones that resulted in less than 2% false positive segmentation in the static frame, and the ones that have higher than 2%. In the total of 793 pairs of videos and motion referring expressions, we find that almost half of the expressions at 380 out of 793, are in the second group, which shows the major concern with the original MeVIS benchmark evaluation vs. our motion-centric evaluation. Furthermore, we prompt GPT-4o with the referred expressions from the two groups

| Dynamic Group | Static Group |
|---|---|
| - Richer in dynamic verb phrases
- Describes multi-step actions
- Shows how actions unfold in context
- Captures transitions and directional movement | - More abstract or static at times
- Static poses
- Less about sequences, more about simple states or high-level summaries |

Table 3: Differentiating the properties of the two major groups of referring expressions based on analysing the false positives in the static frame. The first group has less than 2% false positives in the static keyframe and as such we refer to as the Dynamic group. While the second group has more than 2% false positives and is referred to as the Static group. The differences are automatically generated using GPT-4o by parsing the referring expressions from each group.

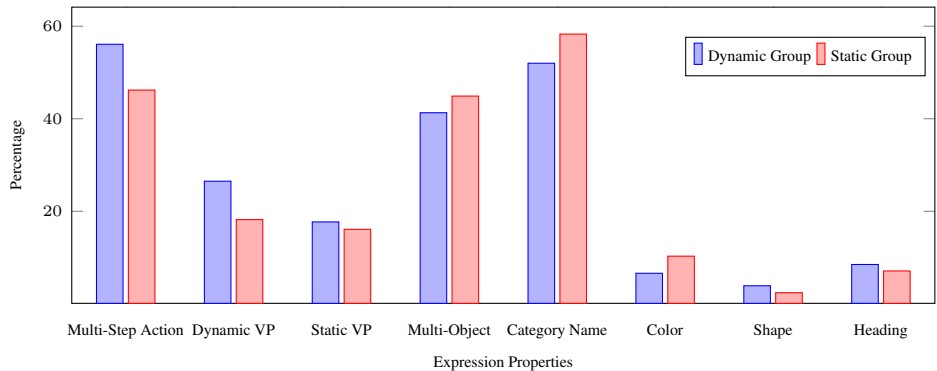

Figure 6: Fine-grained analysis on the properties of the two major groups of referring expressions which are the Dynamic and Static groups. VP: verb phrase.

and inquire "Which group captures better motion actions?". The answer includes a differentiation between the two groups' characteristics as summarized in Table 3 and a correct identification that Group 1 (i.e., Dynamic Group), captures motion actions better.

We take another step to study both the dynamic and static cues conveyed from the referred motion expressions from the identified two groups. Towards that, we use GPT-4o to identify eight main properties in each of the referring expressions through prompting it with the following: (i) "Does the following expression have a multi-step action: <EXP>?", (ii) "Does the following expression have a multi-object interaction: <EXP>?", (iii) "Does the following expression have a rich dynamic verb phrase: <EXP>?", (iv) "Does the following expression include color: <EXP>?", (v) "Does the following expression include shape: <EXP>?", (vi) "Does the following expression describe heading or direction: <EXP>?", (vii) "Does the following expression have a verb indicating static position: <EXP>?" and (viii) "Does the following expression have the subject as an identifiable category: <EXP>?". We show the percentage of expressions within each group that received a response "yes" from GPT-4o for the previous properties, highlighting the differences between both the dynamic and static groups in Figure 6. It shows that the dynamic group is mainly differentiated from the static one with multi-step actions and more dynamic verb phrases. While the static verb phrases in the static group are on par with the dynamic ones. On the other hand, three main properties differentiate the static vs. dynamic group of referred expressions, which are the multi-object interactions, the use of the category name and the color. Such properties give static cues in the referred expression that suffice to use a single image to segment the object.

## 5   Conclusion

In conclusion, we have shown the shortcomings in both referring video segmentation methods and the benchmarks used for their evaluation. We propose a novel benchmark that is motion-centric through the use of a static keyframe paired with the dynamic video to mislead the referring segmentation methods into the existence of the object without motion. Additionally, we propose three strong baselines that outperform the state of the art while being static biased.

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

# A  Additional Implementation Details

For our three baselines, we use the following weights for Qwen2.5-VL that are available from Hugging Face "Qwen2.5-VL-7B-Instruct". For the motion-centric probing and to avoid any bias in our strong baselines in predicting the segmentation tied to a certain location in the image, we rather prompt the models with two versions of the same video; one that has the static key-frame on the left side and another on the right side. Then we use the combination of both predictions for the final evaluation. We found this to be a better evaluation of their capabilities in identifying the real motion referring expression from the fake motion in the static keyframe. Throughout all the experiments, we use an A6000 GPU to run the evaluation of all the models discussed.

# B  Limitations

Our work has limitations tied to evaluating video multi-modal large language models that are conducting spatiotemporal referring segmentation. Such models are GPU memory hungry and require specialized GPUs for inference, let alone their training. Consequently, it limits the contributors to the benchmarks and developing better models that overcome these issues with a focus on motion-centric evaluation, where low-resourced communities who do not have access to such resources can find it impossible to participate in that kind of research.

# C  Impact Statement

Video multi-modal large language models are widely used in various applications, such as robotics, medical image processing and is even useful in temporal imagery in remote sensing. The pixel-level understanding within such MLLMs is necessary for such applications that require the localization and even in certain scenarios, the delineation of the boundaries for the objects of interest. It is even more important to maintain a good spatiotemporal understanding to capture motion and dynamics in the input video. In our work, we have investigated the shortcomings of video MLLMs in the video referring segmentation task, while providing a more challenging motion-centric benchmark to push these models into a better understanding of the temporal information.

However, as with many other AI advancements, there are risks that could be entailed from the deployment of such models. There could be inherent biases emerging in such video MLLMs, impacting various underrepresented groups. We think that our benchmarking efforts, probing and providing a tool to understand the pitfalls in the understanding and reasoning of these models could be an initial direction for mitigating such biases. Nonetheless, we leave it for future work to explore this further.

