# OpenReview forum: "Are we using Motion in Referring Segmentation? A Motion-Centric Evaluation"
_NeurIPS.cc/2025/Conference — Submitted to NeurIPS 2025_

### Official Review · Reviewer_w9oT · 2025-07-01

**Clarity:** 2
**Significance:** 2
**Originality:** 2
**Rating:** 3
**Confidence:** 3

**Summary:**

The paper questions whether current video-language models for referring video segmentation actually use motion cues. To test this, the authors propose a Motion-Centric Evaluation Framework that appends a static keyframe next to each frame in the video, aiming to fool models into making predictions based on static appearance rather than motion.

The paper also introduces three baselines using MLLMs and the Segment Anything Model (SAM). The first baseline runs the MLLM and SAM independently on each frame. The second uses the MLLM to identify the object in the first frame and then tracks it using SAM 2.0. The third improves upon this by using the MLLM to select a "keyframe" where the referring expression is most relevant, initializes tracking there, and propagates it through the video. These baselines perform better than the prior methods.

**Questions:**

- Do you actually concatenate the static keyframe side-by-side with every video frame, or is it added as a separate frame at the start or end? It looks like spatial concatenation from the figures, but the paper never clearly says this. Also, how exactly are predictions combined when you try both keyframe-left and keyframe-right?

Did you try any ablations to rule out the effect of aspect ratio distortion or visual layout change? For example, what happens if you add a blank or unrelated frame? Or if you resize the input back to the original aspect ratio after concatenation? Even inserting the keyframe only at the start / end of the video could help clarify whether the performance drop is due to motion insensitivity or just input format changes.

**Ethical Concerns:**

["NO or VERY MINOR ethics concerns only"]

**Final Justification:**

No score changes. The authors provided some additional information, but it was not sufficient for acceptance. For what it's worth, they presented a good plan for a future submission.

**Limitations:**

Yes.

**Quality:**

3

**Strengths And Weaknesses:**

Strengths;

- The paper raises an important and under-explored question, whether models performing referring video segmentation actually use motion or just rely on static visual cues. This is a meaningful direction, especially as video MLLMs become more widely used across tasks that implicitly assume spatiotemporal reasoning.
- The use of MLLMs in combination with specialist like SAM and SAM 2.0 leads to strong performance than the prior end-to-end models. These baselines serve as a useful reference point for any future study.


Weaknesses
- Evaluation protocol is underspecified, relying on visual inference rather than clear description: The Motion-Centric Evaluation Framework is not clearly described in the main paper or appendix. It appears, based on Figures 3, 4, and 5 that the authors spatially concatenate a static keyframe (left or right) to each frame of the original video. However, this is never explicitly stated in the text. Key details such as how predictions are aggregated across the “keyframe-left” and “keyframe-right” variants, or how models handle the altered input dimensions, are not specified. This limits reproducibility and makes it difficult to confidently interpret the results.

- No ablations to disentangle motion reasoning from visual layout bias: The key claim is that appending a static frame misleads models that lack motion reasoning. However, no baseline is provided to test whether the performance drop is due to motion-related failure or simply a result of distorted input format. For instance, what happens if a black or unrelated frame is appended instead of a keyframe? Without such control experiments, it's entirely possible that the drop in performance is due to the altered aspect ratio or visual duplication rather than any failure in modeling motion.

- Baselines succeed through component strength rather than novel methodology: The three proposed baselines rely on powerful existing tools, Qwen2.5-VL for language-based grounding and SAM 2.0 for tracking. Their superior performance compared to prior work is largely due to the strength of these modules rather than any architectural or algorithmic contribution from the authors. For example, in Baseline 2, the model identifies the object once and tracks it with SAM 2.0, with no further temporal reasoning or refinement. While this is useful to expose weaknesses in end-to-end models, it is not itself a substantial contribution.

---

> ### Author Rebuttal · Authors · 2025-07-25
>
> Response provided under Reviewer 1 comment.

---

### Official Review · Reviewer_Y8AC · 2025-07-01

**Clarity:** 3
**Significance:** 3
**Originality:** 3
**Rating:** 3
**Confidence:** 4

**Summary:**

The paper presents a study exploring how existing MLLMs tend to over-rely on specific frames in video segments to perform video-related dense prediction tasks in existing video understanding benchmark datasets, rather than explicitly learning motion cues. The authors demonstrate this phenomenon through a motion-centric probing, where they append a specific keyframe of the queried motion in a video at one of the temporal edges of the video (at the start or at the end), and then test the performance of MLLMs on these keyframe-appended videos. They show that the video understanding performance of MLLMs on existing benchmarks drops by almost half when using the keyframe-appended videos, indicating an over-reliance on the keyframes and a failure to understand the actual motion cues in the videos. To overcome these shortcomings, the authors propose a baseline that, given a queried motion, leverages the relevant object bounding boxes tracked across the video to perform the dense prediction tasks. While the authors show improved performance on existing benchmarks with their proposed baseline, they also show its continued dependence of MLLMs on motion keyframes rather than entire motion cues, and advocate the need for more motion-centric benchmarks.

**Questions:**

1. Have the authors explored any human studies to ground the performance of Qwen2.5-VL in selecting motion segments from videos (Lines 166-168)? Such grounding can help in understanding the reliability of the MLLM performance, and, by extension, the reliability of the benchmark itself.

2. Have the authors looked at the additional latency of the proposed baselines (last three rows of Table 1) as a function of the number of motions detected and tracked?

3. Based on the distribution of different expression properties for the static and the dynamic groups of motion expressions (Figure 6), is it possible to work towards a necessary and a sufficient set of questions to determine static vs. dynamic motion expression?

**Ethical Concerns:**

["NO or VERY MINOR ethics concerns only"]

**Final Justification:**

While the authors' rebuttal addresses some questions raised in the review, it lacks responses to other specific questions and concerns. Moreover, the suggested changes in the paper from the authors are non-trivial and would require further review. Consequently, the paper is currently below the acceptance threshold in my revised assessment.

**Limitations:**

1. The authors have not mentioned any limitations of choosing the middle frame of a video segment containing a queried motion as the keyframe for the motion in their motion-centric probing experiments.

2. The authors have also not mentioned whether their motion-centric probing approach is limited by different appearances of object motions and different camera motions.

**Quality:**

2

**Strengths And Weaknesses:**

### Strengths
1. The proposed study highlights a specific performance pattern of MLLMs, which provides a clear research direction for a closer inspection at the workings of MLLMs and the design of benchmarks for video understanding.

2. The proposed baseline of augmenting tracked object bounding boxes to the context of MLLMs is sound and is demonstrated to improve performance on various dense prediction tasks in video understanding.

3. The proposed motion-centric benchmark for more thorough testing of the MLLM's ability to understand specific motion cues provides the insightful results, making it useful for further investigations of MLLMs and design of future benchmarks.


### Weaknesses
1. It is unclear how generalizable the proposed motion-centric probing approach is to different appearances of motions and other data capturing artifacts in videos. For example, is the middle frame of the motion segment the optimal choice for the keyframe regardless of how the motion appears (e.g., if the motion is a person walking and the middle frame shows the person to be in an almost standing position in-between foot phases)? Does the middle frame continue to remain the most optimal choice if the speed of the motion relative to the camera increases/decreases, and if there are camera motions involved?

2. The choice of the middle frame as the keyframe is also sensitive to the first and last frames chosen by the MLLM used for motion-centric probing. Have the authors tested the robustness of their probing approach to the performance of the MLLM by, for example, querying the MLLM multiple times and taking an average (mean/median/mode) first and last frame from the sets of returned first and last frames? Performing a robustness analysis would provide a complete picture of how sensitive the MLLMs actually are to the motion keyframes.

3. How did the authors arrive at the observation that appending the motion keyframe at one of the temporal edges of the video leads to the MLLMs being misled by the keyframe? Have they experimented with appending the keyframe at other positions in the video? Have they observed MLLMs putting extra attention at the temporal edges of videos?

---

> ### Author Rebuttal · Authors · 2025-07-25
>
> Response provided under Reviewer 1 comment.

---

> ### Comment · Reviewer_Y8AC · 2025-08-07
> **Rebuttal Response**
>
> I thank the authors for their rebuttal. However, the lack of point-to-point response is disappointing (as Reviewer gZzZ also mentioned), and I was unable to find relevant responses to my questions in the rebuttal. Additionally, based on the other reviews and the authors' response, it appears that the paper would require non-trivial revisions, which, in turn, would require further review. Combining these two factors, the paper currently falls below the acceptance threshold in my revised assessment.

---

### Official Review · Reviewer_gZzZ · 2025-07-01

**Clarity:** 2
**Significance:** 2
**Originality:** 2
**Rating:** 2
**Confidence:** 3

**Summary:**

The authors studied the presence of motion in dense video segmentation tasks, particularly (i) examining whether motion is used in referring segmentation and (ii) whether existing MLLMs leverage motion features for segmentation based on natural language expressions. To capture motion-referring expressions, the authors identified that a single frame can often suffice without any temporal reasoning. Addressing this, they introduced a motion-centric probing and evaluation framework to select keyframes within videos that mislead models with apparent motion that lacks true spatiotemporal changes. Their empirical analysis revealed the underutilization of MLLMs in dense prediction tasks, while further establishing new baselines that offer insights into the relationship between spatial and temporal information in dense video-language understanding tasks.

**Questions:**

Please refer to the "Strengths And Weaknesses" section. I tried to summarize my findings and questions there.

**Ethical Concerns:**

["NO or VERY MINOR ethics concerns only"]

**Limitations:**

Yes

**Quality:**

2

**Strengths And Weaknesses:**

**Strengths**

S1. Analyzing the limitations of video MLLMs in utilizing temporal information in dense prediction tasks is interesting.

S2. The evaluation technique for referring video segmentation based on keyframe selection is unique.

S3. The authors identified and studied an intriguing observation that some motion expressions do not require motion in video MLLMs.

**Weaknesses**

*W1. Limited Scope and Core Assumptions.* I found the assumption that the majority of actions or motions can be inferred from one frame to be ambiguous and inappropriate for medium or long-range video understanding. For short videos or image frames, consisting of mostly an equal number of objects that do not vary in background, might support their statement. However, in terms of medium to long-range videos having varying background motion (other than the object motion), will not align with this argument. This is one of the primary limitations that constrains the study to a shorter range of tasks and benchmarks. Furthermore, I believe the authors failed to support their claims that Video MLLMs fail to utilize temporal information without providing systematic empirical evidence. They do not actually test existing video MLLMs on motion-referring tasks.

*W2. Unclear Problem Definition and Missing Justification.* What are the actual limitations of Video MLLMs that the authors wanted to discuss in Sec. 3.1? I believe the authors tried to discuss the properties of motion-referring expressions. But what one might want to know is why and how existing video MLLMs cannot efficiently capture such important information—whether similar experiments have been done or studied. Given this, the goal of this study remains unclear and incomprehensive. Besides, how did the authors come up with the properties of such expression benchmarks?

*W3. Motivation of Probing and its Structure.* The authors might want to thoroughly describe the necessity of the introduced motion-centric probing and how it theoretically or empirically it solves the problem that existing benchmarking cannot.

*W4. Weak Methodology and Inadequate Analysis.* The development of the study along with the probing structure is very weak and unclear. I found that the automatic keyframe selection approach is not well explained. Besides, the authors did not adequately address when temporal information is necessary versus when static information suffices.

**Minor Weaknesses**

- The figures lack proper understanding and context, which I believe are not aligning with the captions. For example, how can we confirm that a single image, a.k.a., the keyframe, is sufficient to capture motion-referring expressions without dynamic information? In the second figure, I would assume it's a typical text-based segmentation—what and how would one qualitatively analyze from this figure? In summary, I would suggest that the authors justify what the finding and necessity of showing such figures to the readers are, focusing on the primary goal of the paper.

- While Fig. 3 shows their "motion-centric benchmark and evaluation" according to the authors, I believe it is actually testing static-centric capability. This leads to the realization that the approach conflates two different problems: whether models use temporal information and whether it is necessary for the task.

While this study presents an interesting idea of investigating motion expressions in MLLMs, the current version lacks clear methodological development, sufficient experiments, and contribution clarity, and has limited novelty. With that being said, this current version of the paper falls below the threshold for NeurIPS. I encourage the authors to continue developing this promising direction.

---

> ### Author Rebuttal · Authors · 2025-07-25
>
> Response provided under Reviewer 1 comment.

---

> > ### Comment · Reviewer_gZzZ · 2025-08-06
> >
> > I want to thank the authors for their summarized responses. While I appreciate their decision not to provide point-to-point responses, the reason given—"Since the majority of the reviews are on the Reject side"—is quite disappointing. If the authors have solid explanations or arguments that could address reviewers' possible confusion or concerns, they should provide them. Offering point-to-point responses only when the paper receives medium/good-level scores is not necessarily justified.
> >
> > Regarding the rebuttal, some of the new experimental findings are interesting and could be included in the revised version. I believe that whether it's a benchmarking or methodological advancement paper, there should be a clear motivation and rationale for why a reader would read the paper and what new or different insights it offers. I would request the authors to focus on this point in their revised version, as it would strengthen the paper.
> >
> > I wish the authors the best of luck!

---

### Official Review · Reviewer_esWL · 2025-07-04

**Clarity:** 2
**Significance:** 3
**Originality:** 3
**Rating:** 2
**Confidence:** 5

**Summary:**

The paper questions whether current video MLLMs truly leverage motion in referring video segmentation. It introduces a motion-centric evaluation that uses misleading static keyframes to test models' reliance on temporal cues. Surprisingly, strong baselines using static frames outperform prior methods, revealing that many models underuse motion. The authors also analyze referring expressions to distinguish those that require genuine temporal understanding, highlighting the need for better spatiotemporal grounding in future models.

**Questions:**

See my weakness.

**Ethical Concerns:**

["NO or VERY MINOR ethics concerns only"]

**Final Justification:**

After reading the rebuttal and other reviewers’ comments, I find that the paper still lacks a strong perspective for analyzing current MLLMs on motion clues, and the writing remains poorly organized. I encourage the authors to thoroughly revise and restructure the paper before resubmitting to a future venue.

**Limitations:**

Yes.

**Paper Formatting Concerns:**

The current citation format appears inconsistent with standard NeurIPS conventions, which typically use numerical references (e.g., [1], [2]) within the main text, corresponding to a numbered reference list at the end. Adopting this style would improve clarity and alignment with common formatting practices.

**Quality:**

2

**Strengths And Weaknesses:**

Strengths:

1. The paper presents an interesting idea and offers in-depth analysis. It effectively highlights the limitations of current multi-modal large language models (MLLMs) in video object segmentation, particularly their underutilization of motion cues.

Weaknesses:

1. The description of the motion-centric probing experiment lacks clarity. Despite careful reading, I could not clearly determine the exact setup used by the authors. For instance, Line 171 mentions “a static keyframe in addition to the original video clip”—is this a spatial augmentation? If it is the case, one possible confounding factor could be resolution. Since the solutions go double, the MLLMs may become confused. A quantitative study to rule out such artifacts would be valuable.

2. The experimental presentation could be better structured. Currently, the logic flow feels scattered. It would be clearer to introduce the motion-centric benchmark first, followed immediately by the relevant results (Table 2), rather than interleaving it with other comparisons (Table 1).

3. The proposed baseline—using an MLLM to identify keyframes and perform grounding, followed by SAM 2 for propagation—lacks technical novelty. This weakens the overall contribution, though it achieves good performance.

4. I have some concerns regarding the paper's interpretation of motion expressions. In practice, even humans often rely on a few discriminative frames to identify the target object, rather than processing the entire temporal sequence. Given that most MLLMs use image-based vision encoders like CLIP, it’s expected that they struggle with motion understanding. Therefore, it may be unrealistic to expect current MLLMs to truly reason about motion, and the observed limitations may reflect architectural constraints rather than conceptual shortcomings.

---

> ### Author Rebuttal · Authors · 2025-07-25
>
> We thank the reviewers for their comments. Since the majority of the reviews are on the Reject side, we opt not to respond point by point as a standard rebuttal, but rather provide one response to few of the major points for all the reviewers and provide details on how we plan to improve the work for future submissions.
>
> A- Our work shows that simple baselines that rely on a single frame or quasi dynamic information that is only on the coarse level not the full dense spatiotemporal information outperform SOA in the video referring segmentation. More importantly, it shows that these baselines, in addition to former methods, fail to differentiate between the static component in the video (i.e., static frame), meant for deceiving these models, from the dynamic component in the video. Quantitative results in Table 2 comparing mini-validation vs. the Motion Centric one, showing a considerable drop in performance, confirm that. Qualitative results also confirm this in Figs 4 & 5. Hence, we show with sufficient evidence that pixel-level video MLLMs in addition to recent video MLLMs coupled with SAM/2.0 such as Qwen2.5-VL fail to capture the full spatiotemporal information to be able to differentiate between true motion and deceiving ones. We use both quantitative and qualitative means! Quoting one of the reviewers "They do not actually test existing video MLLMs on motion-referring tasks." False, we evaluate VideoGLAMM and Qwen2.5-VL these are the best in their respective tasks by the time of the submission. We also add here Sa2VA[1] concurrent unpublished work to show case that our benchmark is definitely needed!
>
> B- While the Why is interesting, yet is not a necessary component of the benchmarking effort, which is the focus of our study. However, for the how, we provide a simple mechanism of comparing concurrent SOA model for video referring segmentation, Sa2VA[1], and we LoRA fine-tune the backbone ViT with MeVIS training set with videos containing deceiving static frames (MeVIS Motion Centric - Train set), in addition to all the other referring video segmentation datasets Sa2VA was training on.
> Results show that on our MeVIS-Motion Centric benchmark (i.e., mini-validation motion centric split) a gain of around 2% is attained, yet there is still room to improve this. Where our goal is to push video MLLMs to truly capture dense spatiotemporal information in the videos and not be impacted by such deceiving probing.
>
> | Method  |  Backbone/Base MLLM  |  J | F  |  J&F |
> |---|---|---|---|---|
> | VideoGLAMM  | VEn+Phi3-Mini-3.8B  | 18.1  | 25.1  | 21.6  |
> | MLLM+SAM2.0+Keyframe  |  Qwen2.5-VL-7B | 23.4  |  34.2 |  28.7 |
> |  SA2VA [1]  | InternVL2.5-8B  |  23.5 | 33.5  | 28.5  |
> |  SA2VA [1] - MoCentric |  InternVL2.5-8B  |  **25.0**   | **35.3**  | **30.1** |
>
>
> **Minor Fix was added to the last row from a fixed experiment**
>
> 1- Yuan, Haobo, et al. "Sa2va: Marrying sam2 with llava for dense grounded understanding of images and videos." arXiv preprint arXiv:2501.04001 (2025).
>
> C- We find it unnecessary to motivate the reason why we need to capture true motion in Video MLLMs and differentiate deceiving static frames from dynamic ones since the literature is too huge on that aspect and it is a well-established norm that we push models to capture true dynamic information; we refer to a few here. **All of these focus on MLLMs for video Q/A and do not focus on pixel-level visual grounding or referring, unlike ours. They are concurrent works to ours, and the first 2 are not published yet**:
>
> 1- Xue, Zihui, Mi Luo, and Kristen Grauman. "Seeing the Arrow of Time in Large Multimodal Models." arXiv preprint arXiv:2506.03340 (2025).
>
> 2- Cores, Daniel, et al. "Lost in Time: A New Temporal Benchmark for VideoLLMs." arXiv preprint arXiv:2410.07752 (2024).
>
> 3- Zohar, Orr, et al. "Apollo: An exploration of video understanding in large multimodal models." Proceedings of the Computer Vision and Pattern Recognition Conference. 2025.
>
> Nonetheless, the easiest scenario to motivate the above is in robotics. The difference between a pedestrian standing on the sidewalk with a heading showing it is about to cross the street vs. a pedestrian that is moving and crossing the street. The difference between a static robotic arm that has the object grasped and awaiting the human to catch it from the robot hand vs. a robotic arm grasping the object and moving towards the human hand. **In various human-2-robot and robot-2-robot interaction settings, a full understanding of the true motion and its existence with grounding is crucial for safety.**
>
> D- We truly appreciate the reviewer that pointed to the spatial bias that could be a confounding factor indeed its a good comment. Except, we already take that into consideration referring to the Appendix where all our baselines evaluate on both the static frame on the left side and the static frame on the right side to avoid such bias. Nonetheless, we can expand with more ablations on this.
>
> We plan to include the following in future submissions:
>
> 1- Better ablations on the selected frame for deceiving these models (first/last vs. keyframe)
>
> 2- Better ablations on the spatial bias from the use of two concatenated frames, and if it is a confounding factor w.r.t capturing static vs. dynamic components in the video or not.
>
> 3- Our work currently focuses on differentiating dynamic multi-frame component vs. one static frame, which we refer to as **Motion Existence**. We plan to extend it to study how the order of multi-frame component can impact the referring segmentation, which we refer to as **Motion Order**. Both are important to study within pixel-level visual grounding.
>
> 4- Finally, to better motivate the need for motion, we plan to build a comprehensive evaluation benchmark for pixel-level visual grounding focused on motion-order and motion-existence probing (i.e., that is motion-centric) and is out of distribution from the current training data used in various video MLLMs with focus on three robotics-relevant settings: (autonomous driving, robot manipulation and egocentric human manipulation).

---

> > ### Comment · Reviewer_esWL · 2025-08-05
> >
> > I appreciate the authors' efforts during the rebuttal, which addressed some of my concerns, such as the baseline setting. However, I believe the paper is still below the acceptance threshold and requires substantial revisions.

---

### Decision · Program_Chairs · 2025-09-17

**Decision:**

Reject

**Comment:**

Reviewers raised several concerns about novelty, clarity, experiments, and analyses. The authors acknowledged the concerns and provided a consolidated feedback outlining how they plan to improve their future submission. A little unconventional, but the reviewers still appreciated. Given the lack of thorough response in the rebuttal, the reviewers maintained their original ratings, leading to an overall rejection recommendation. This meta reviewer did not find significant reasons to overturn their recommendations, hence recommends rejection at this time.